# Prognosis after resection of single large hepatocellular carcinoma: Results from an Asian high-volume liver surgery center

**Yueh-Wei Liu[1], Chih-Che Lin[1], Chee-Chien Yong[1], Chih-Chi Wang[1], Chao-Long Chen[1], Jing-Houng Wang[2], Yi-Hao Yen[2]***

1 Liver Transplantation Center and Department of Surgery, Kaohsiung Chang Gung Memorial Hospital, Kaohsiung, Taiwan, 2 Division of Hepatogastroenterology, Department of Internal Medicine, Kaohsiung Chang Gung Memorial Hospital and Chang Gung University College of Medicine, Kaohsiung, Taiwan

* cassellyen@yahoo.com.tw

**Data Availability Statement:** All relevant data are within the paper and its Supporting Information files.

## Abstract

### Background & aims

In 2012, the Barcelona Clinic Liver Cancer (BCLC) system designated a single large hepatocellular carcinoma (SLHCC) (>5 cm) as BCLC stage A rather than stage B. However, a recent study from western countries reported that prognosis following liver resection (LR) among patients with SLHCC was similar to that among patients with BCLC stage B. We aim to evaluate the prognosis following LR among patients with SLHCC from an Asian high-volume liver surgery center.

### Methods

Patients who underwent curative-intent LR for histologically proven HCC between 2011 and 2017 were enrolled using an HCC registry database. Overall survival (OS) among patients with BCLC stages 0, A, and B was examined. Patients with a SLHCC were classified as BCLC stage A1.

### Results

Among 543 patients, 89 (16.4%) were BCLC stage 0, 289 (53.2%) were BCLC stage A, 92 (16.9%) were BCLC stage A1, and 73 (13.4%) were BCLC stage B. The median follow-up was 38 months. The five-year OS rates among patients with BCLC stages 0, A, A1, and B were 83.5%, 83.7%, 77.4%, and 55.4%, respectively (p<0.001). No difference in OS was noted for patients with BCLC stage A versus A1 (p = 0.11), even after adjusting for competing factors (hazard ratio = 0.97, 95% confidence interval = 0.53–1.79; p = 0.93).

### Conclusion

Prognosis following LR among patients with SLHCC was similar to that among patients with BCLC stage A. The prognosis for SLHCC should thus be considered comparable to that for BCLC stage A.

**Funding:** This study was supported by Grant CMRPG8J1281 from the Kaohsiung Chang Gung Memorial Hospital, Taiwan. Grant Recipient is Yi-Hao Yen. The funders had no role in study design, data collection and analysis, decision to publish, or preparation of the manuscript.

**Competing interests:** The authors have declared that no competing interests exist.

## Introduction

Hepatocellular carcinoma (HCC) is one of the leading cause of cancer death worldwide [1]. Chronic viral infection accounts for the majority of HCC etiologies in Taiwan, where around 1.70 million subjects have hepatitis C virus (HCV) infections and around 3.50 million people are hepatitis B virus (HBV) carriers [2]. HCC and viral hepatitis are among the major public health threats in Taiwan.

Liver resection (LR) improves the long-term survival of patients with HCC across different Barcelona Clinic Liver Cancer (BCLC) stages [3]. The BCLC system designated a single large HCC (>5 cm) as BCLC stage A rather than stage B in 2012 [4]. The revised BCLC classification schema has been endorsed by the American Association for the Study of Liver Diseases (AASLD) [5] and the European Association for the Study of the Liver (EASL) [6].

In practice, tumors >5 cm are indicated for LR as early HCC (BCLC-A), but appear to bear a worse prognosis than HCC within Milan criteria [4]. Some authors have designated this sub-group as BCLC-AB stage [3]. Furthermore, a recent international multi-institutional study from western countries reported that prognosis following LR among patients with single large HCC was similar to that among patients presenting with BCLC-B HCCs [7].

Given the debate regarding the prognostic stratification of single large HCC, we sought to evaluate the outcomes of patients undergoing LR for BCLC stages 0, A, and B, with a particular focus on patients with a single large HCC.

## Patients and methods

### Patients

The data used in this study were extracted from the Kaohsiung Chang Gung Memorial Hospital HCC registry database. Five hundred and forty-three naïve HCC patients with BCLC stages 0, A, and B who received LR with curative-intent from January 2011 to December 2017 at Kaohsiung Chang Gung Memorial hospital were enrolled. A flow chart of the patients' enrollment is shown in Fig 1. We checked the vital status of these patients using the Cancers

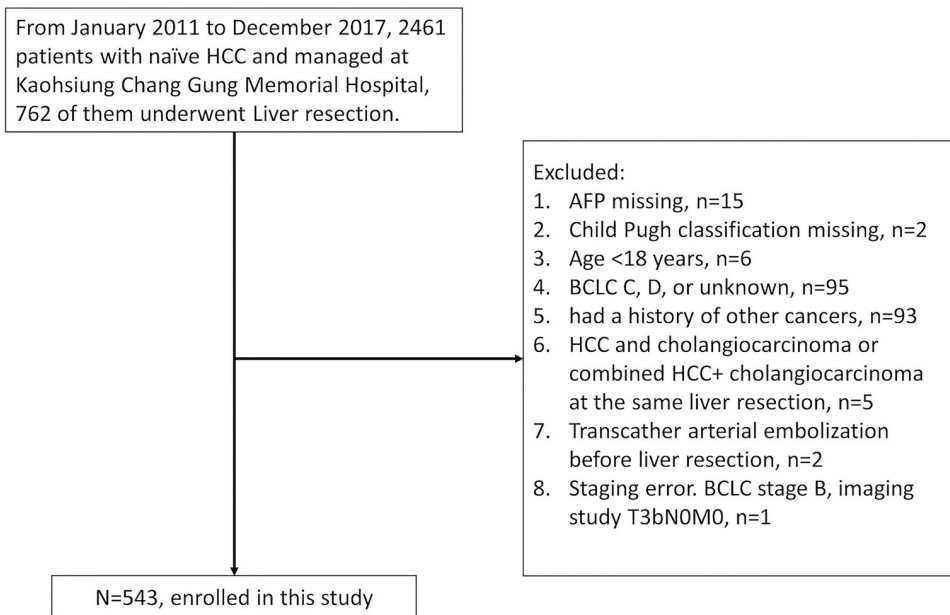

**Fig 1. Flow chart of the patient enrollment from the Kaohsiung Chang Gung Memorial Hospital (KCGMH) cancer database.**

Screening and Tracing Information Integrated System for Taiwan (https://hosplab.hpa.gov.tw/CSTIIS/index.aspx). All the procedures used in the study were in accordance with the ethical standards of the committees on human experimentation and with the Helsinki Declaration. This study was approved by the Institutional Review Board of Chang Gung Memorial Hospital (IRB number: 201901892B0). The requirement for informed consent was waived by the IRB.

### Variables and outcomes of interest

The primary outcome was overall survival (OS), which was defined as the time interval between the date of LR and the date of death or last follow-up. The BCLC stages were defined by the EASL guideline [4]. For the purpose of this study, patients with a single large HCC (>5 cm) were classified as BCLC stage A1. Patients with HCC within Milan criteria [4] were classified as BCLC stage A.

The demographic and clinical characteristics collected included age, gender, cirrhosis status (defined by Ishak score 5 or 6) [8], serum hepatitis B surface antigen (HBsAg) or anti-HCV positivity, body mass index (BMI), blood test results (i.e. a-fetoprotein [AFP] and total bilirubin levels, international normalized ratio [INR], Child–Pugh liver function classification), BCLC staging, type of surgical resection (i.e. major or minor), tumor differentiation (Edmondson-Steiner grade)[9], maximal tumor size (according to the postoperative pathological examinations), pathologic microvascular invasion and microsatellite nodule status, and resection margin status (i.e. R0 or non-R0). Major resection was defined as resection of three or more Couinaud segments.

### Statistical analysis

The demographic and clinical characteristics of the entire cohort were summarized as median (interquartile range [IQR]) and frequency (%) for continuous and categorical variables, respectively. Cox proportional hazard regressions were conducted to evaluate the impact of risk factors on all-cause mortality using univariate and multivariate analyses. The variables with $P$ values less than 0.05 in univariate analysis were included in multivariate model. Median OS months with 95% confidence intervals (CIs) for each BCLC stage were computed using Kaplan-Meier analysis. The survival curve of each BCLC stage was illustrated using the Kaplan-Meier estimator, and the survival differences between groups were estimated using the log-rank test. The pairwise comparisons between each pair of BCLC stages were also estimated using the log-rank test. The statistical significance level was set at $P$ values less than 0.05. All statistical analyses were performed using Stata version 14.0 (StataCorp. 2015. Stata Statistical Software: Release 14. College Station, TX: StataCorp LP).

## Results

### Patient and tumor characteristics

The present cohort was comprised of 543 patients with BCLC stages 0, A, and B HCC who underwent LR. The median follow-up was 38 months (IQR 26–63). Patients were categorized into the BCLC 0 (n = 89, 16.4%), BCLC A (n = 289, 53.2%), BCLC A1 (n = 92, 16.9%), and BCLC B (n = 73, 13.4%) stages (Table 1). The age, gender, BMI, and proportions of patients with HBsAg positive status, total bilirubin ≤1.2 mg/dL, INR≤1.1, Child-Pugh class A, and tumor differentiation were not significantly different between groups. The proportion of patients without cirrhosis (78.3%) was highest (p<0.001), the proportion of patients with anti-HCV positive status (22.8%) was lowest (p = 0.03), the proportion of patients with non-HBV,

**Table 1. Demographics and patient characteristics in the entire cohort (n = 543).**

| Variables | BCLC staging classification | | | | P |
|---|---|---|---|---|---|
| | 0 (n = 89) | A (n = 289) | A1 (n = 92) | B (n = 73) | |
| Age, years [median (IQR)] | 60 (52–63) | 61 (54–68) | 63 (53.5–71) | 60 (53–67) | 0.12 |
| <= 65 | 69 (77.5) | 201 (69.6) | 54 (58.7) | 52 (71.2) | **0.05** |
| >65 | 20 (22.5) | 88 (30.4) | 38 (41.3) | 21 (28.8) | |
| Sex | | | | | 0.09 |
| Male | 61 (68.5) | 232 (80.3) | 75 (81.5) | 59 (80.8) | |
| Female | 28 (31.5) | 57 (19.7) | 17 (18.5) | 14 (19.2) | |
| Cirrhosis | | | | | **<0.001** |
| No | 36 (40.4) | 164 (56.7) | 72 (78.3) | 44 (60.3) | |
| Yes | 53 (59.6) | 125 (43.3) | 20 (21.7) | 29 (39.7) | |
| HBsAg positive | | | | | 0.59 |
| No | 40 (44.9) | 139 (48.1) | 47 (51.1) | 30 (41.1) | |
| Yes | 49 (55.1) | 150 (51.9) | 45 (48.9) | 43 (58.9) | |
| Anti-HCV positive | | | | | **0.03** |
| No | 51 (57.3) | 185 (64.0) | 71 (77.2) | 51 (69.9) | |
| Yes | 38 (42.7) | 104 (36.0) | 21 (22.8) | 22 (30.1) | |
| Alcohol use disorder | | | | | 0.17 |
| No | 83 (93.3) | 276 (95.5) | 84 (91.3) | 65 (89) | |
| Yes | 6 (6.7) | 13 (4.5) | 8 (8.7) | 8 (11) | |
| Non-HBV, non-HCV, non-alcohol use disorder | | | | | **0.002** |
| No | 83 (93.3) | 248 (85.8) | 67 (72.8) | 61 (83.6) | |
| Yes | 6 (6.7) | 41 (14.2) | 25 (27.2) | 12 (16.4) | |
| BMI (kg/m$^2$) | | | | | 0.21 |
| <18 | 2 (2.2) | 4 (1.4) | 2 (2.2) | 2 (2.7) | |
| 18–24 | 37 (41.6) | 113 (39.1) | 43 (46.7) | 39 (53.4) | |
| >24 | 49 (55.1) | 169 (58.5) | 44 (47.8) | 32 (43.8) | |
| Unknown | 1 (1.1) | 3 (1.0) | 3 (3.3) | - | |
| Bilirubin (mg/dL) | | | | | 0.61 |
| <= 12 | 71 (79.8) | 240 (83.0) | 76 (82.6) | 56 (76.7) | |
| >12 | 18 (20.2) | 49 (17.0) | 16 (17.4) | 17 (23.3) | |
| INR | | | | | 0.26 |
| <= 11 | 88 (98.9) | 272 (94.1) | 89 (96.7) | 69 (94.5) | |
| >11 | 1 (1.1) | 17 (5.9) | 3 (3.3) | 4 (5.5) | |
| AFP (ng/dL) | | | | | **<0.001** |
| <= 40 | 81 (91.0) | 249 (86.2) | 73 (79.3) | 47 (64.4) | |
| >40 | 8 (9.0) | 40 (13.8) | 19 (20.7) | 26 (35.6) | |
| Child–Pugh classification | | | | | 0.41 |
| A | 89 (100) | 284 (98.3) | 89 (96.7) | 72 (98.6) | |
| B | - | 5 (1.7) | 3 (3.3) | 1 (1.4) | |
| Type of resection | | | | | **<0.001** |
| Minor | 68 (76.4) | 167 (57.8) | 24 (26.1) | 20 (27.4) | |
| Major | 21 (23.6) | 122 (42.2) | 68 (73.9) | 53 (72.6) | |
| Tumor size, mm [median (IQR)] | 16 (14–20) | 29 (23–35) | 66 (55–90) | 58 (42–84) | **<0.001** |
| Tumor Differentiation | | | | | 0.84 |
| Well or moderate | 85 (95.5) | 274 (94.8) | 90 (97.8) | 69 (94.5) | |
| Poor | 3 (3.4) | 13 (4.5) | 2 (2.2) | 4 (5.5) | |
| Unknown | 1 (1.1) | 2 (0.7) | - | - | |

(*Continued*)

**Table 1.** (Continued)

| Variables | BCLC staging classification | | | | *P* |
|---|---|---|---|---|---|
| | **0 (n = 89)** | **A (n = 289)** | **A1 (n = 92)** | **B (n = 73)** | |
| Microvascular invasion | | | | | **<0.001** |
| No | 58 (65.2) | 128 (44.3) | 31 (33.7) | 26 (35.6) | |
| Yes | 31 (34.8) | 161 (55.7) | 61 (66.3) | 47 (64.4) | |
| Microsatellites nodules | | | | | **<0.001** |
| No | 87 (97.8) | 284 (98.3) | 82 (89.1) | 53 (72.6) | |
| Yes | 2 (2.2) | 5 (1.7) | 10 (10.9) | 20 (27.4) | |
| R0 resection | | | | | **0.03** |
| No | 1 (1.1) | 4 (1.4) | 4 (4.3) | 5 (6.8) | |
| Yes | 88 (98.9) | 285 (98.6) | 88 (95.7) | 68 (93.2) | |

HBsAg, hepatitis B surface antigen; HCV, hepatitis C virus; INR, International Normalized Ratio; AFP, Alpha fetoprotein; BCLC, Barcelona Clinic Liver Cancer; BMI, body mass index; HBV, hepatitis B virus

non-HCV, non-alcohol use disorder (27.2%) was highest (p = 0.002), the proportion of patients with major resection (73.9%) was highest (p<0.001), the proportion of patients with microvascular invasion (66.3%) was highest in the BCLC A1 group (p<0.001), and the median tumor size was largest (6.6 cm) in the BCLC A1 group (p<0.001).

## Survival analysis: BCLC stages 0, A, A1, and B HCC

The 5-year OS rates among the patients with BCLC stages 0, A, A1, and B HCC were 83.5%, 83.7%, 77.4%, and 55.4%, respectively (p<0.001) (Fig 2). There were no significant differences in OS among the patients with BCLC stage 0 versus A (p = 0.28) or BCLC stage A versus A1

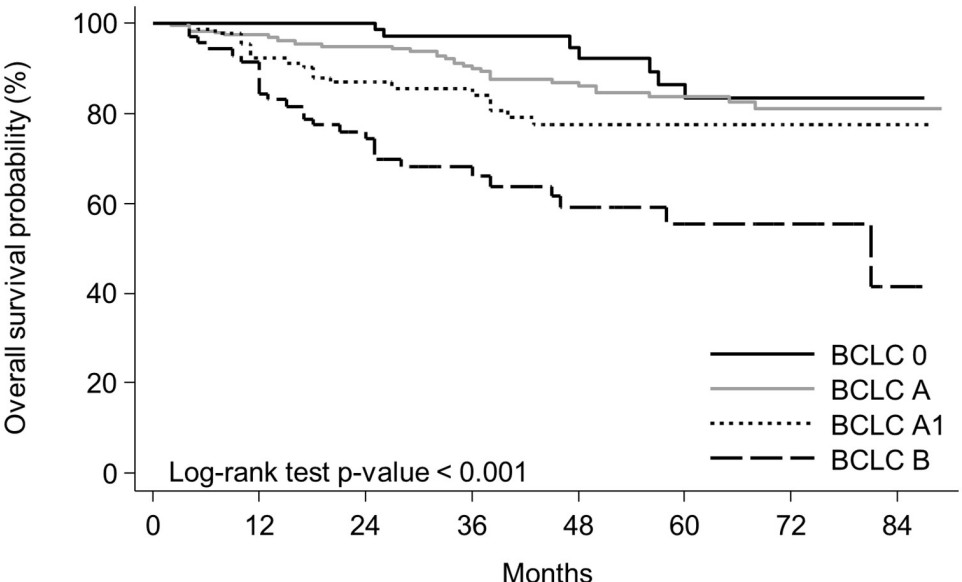

**Fig 2. Survival of patients according to BCLC subclasses.** 5-year overall survival (OS) among patients with BCLC stage 0, A, A1, and B HCC was 83.5%, 83.7%, 77.4%, and 55.4%, respectively (p<0.001). BCLC 0 vs A, p = 0.28, BCLC 0 vs A1, p = 0.03, BCLC 0 vs B, p < 0.001, BCLC A vs A1, p = 0.11, BCLC A vs B, p < 0.001, BCLC A1 vs B, p = 0.005.

(p = 0.11). There was a significant difference in OS among the patients with BCLC stage A1 versus B (p = 0.005).

## Univariate analysis for mortality

The univariate analysis results for mortality are shown in Table 2. Age >65 years [hazard ratio (HR) = 1.98, 95%CI = 1.30–3.02, p = 0.001], total bilirubin >1.2 mg/dL (HR = 1.67, 95% CI = 1.04–2.69, p = 0.03), AFP>400 ng/dL (HR = 2.0, 95%CI = 1.24–3.22, p = 0.004), BCLC stage B [(using BCLC stage A as reference), HR = 3.78, 95%CI = 2.30–6.22, p<0.001], major resection (HR = 1.95, 95%CI = 1.26–3.01, p = 0.003), poor tumor differentiation (HR = 3.75, 95%CI = 1.87–7.53, p<0.001), microscopic vascular invasion (HR = 2.60, 95%CI = 1.6–4.21, p<0.001), microsatellite nodules (HR = 6.88, 95%CI = 4.23–11.2, p<0.001), and without R0 resection (HR = 5.31, 95%CI = 2.44–11.52, p<0.01) were associated with mortality.

## Multivariate analysis for mortality

The multivariate analysis results for mortality are shown in Table 3. Age>65 years (HR = 2.14, 95%CI = 1.37–3.33, p = 0.001), BCLC stage B [(using BCLC stage A as reference), HR = 2.10, 95%CI = 1.16–3.78, p = 0.014], poor tumor differentiation (HR = 2.60, 95%CI = 1.26–5.35, p = 0.01), microvascular invasion (HR = 2.00, 95%CI = 1.21–3.29, p = 0.007), and microsatellite nodules (HR = 3.73, 95%CI = 2.09–6.64, P<0.001) were independent risk factors for mortality.

## Discussion

Although the BCLC staging system has been adopted worldwide, there has been skepticism regarding whether the current BCLC classification performs well in terms of prognostic stratification, especially for patients with a single large HCC, which was previously categorized as BCLC stage B [10–12]. However, the patients enrolled in previous studies [10–12] received treatments other than LR, including transcatheter arterial chemoembolization (TACE) and radiofrequency ablation (RFA). Patients who received local regional therapy other than LR may have had inadequate liver function reserve or severe comorbidities that were contraindications for LR. Therefore, it is not appropriate to compare the prognosis of patients with single large HCC and the prognosis of BCLC B patients and patients within Milan criteria in a cohort enrolling patients who received treatments other than LR.

A recent international multi-institutional study from western countries reported that prognosis following LR among patients with single large HCC was similar to that among patients with BCLC stage B [7]. In contrast, prognosis following LR among patients with single large HCC was similar to that among patients with BCLC stage A in our study. The discrepancy between the results reported by Tsilimigras et al. [7] and those of our study could be due to the single large HCC and BCLC stage B patients being heterogeneous. Furthermore, the numbers of single large HCC and BCLC stage B cases were limited in our study. However, the most important point in the survival analysis is how to make sure the vital status of patients who were lost of follow up. In the 'Patients and methods' section, Tsilimigras et al. did not mention how to check the vital status of patients who were lost of follow up [7]. Theoretically, these patients were categorized as having a censored event (defined as "0"). If the mortality [which were categorized as having an endpoint event (defined as "1")] case number is significant, the result of their study could be different.

No details of the clinical-pathological characteristics of the single large HCC patients included in the Tsilimigras et al. study were provided in that study [7]. In our single large HCC cohort, 21.7% of the patients were cirrhotic, 20.7% of the patients had AFP>400 ng/dL,

**Table 2. Risk factors for mortality (univariate analysis).**

| Variables | HR (95% CI) | P |
|---|:---:|:---:|
| Age, years | | |
| <= 65 | ref | |
| >65 | 1.98 (1.3–3.02) | **0.001** |
| Sex | | |
| Male | ref | |
| Female | 0.66 (0.37–1.16) | 0.15 |
| Cirrhosis | | |
| No | ref | |
| Yes | 1.06 (0.69–1.61) | 0.80 |
| HBs Ag positive | | |
| No | ref | |
| Yes | 0.83 (0.55–1.27) | 0.40 |
| Anti-HCV positive | | |
| No | ref | |
| Yes | 1.23 (0.8–1.9) | 0.34 |
| BMI (kg/m$^2$) | | |
| <18 or 18–24 | ref | |
| >24 | 0.98 (0.65–1.49) | 0.93 |
| Bilirubin (mg/dL) | | |
| <= 1.2 | ref | |
| >1.2 | 1.67 (1.04–2.69) | **0.03** |
| INR | | |
| <= 1.1 | ref | |
| >1.1 | 0.6 (0.19–1.9) | 0.38 |
| AFP (ng/dL) | | |
| <= 400 | ref | |
| >400 | 2 (1.24–3.22) | **0.004** |
| Child–Pugh classification | | |
| A | ref | |
| B | 1.12 (0.27–4.55) | 0.88 |
| BCLC staging classification | | |
| 0 | 0.64 (0.28–1.44) | 0.28 |
| A | ref | |
| A1 | 1.60 (0.91–2.82) | 0.11 |
| B | 3.78 (2.30–6.22) | <**0.001** |
| Type of resection | | |
| Minor | ref | |
| Major | 1.95 (1.26–3.01) | **0.003** |
| Tumor Differentiation (Edmondson-Steiner grade) | | |
| Well or moderate | ref | |
| Poor | 3.75 (1.87–7.53) | <**0.001** |
| Microvascular invasion | | |
| No | ref | |
| Yes | 2.6 (1.6–4.21) | <**0.001** |
| Microsatellites nodules | | |
| No | ref | |
| Yes | 6.88 (4.23–11.2) | <**0.001** |

(*Continued*)

**Table 2.** (Continued)

| Variables | HR (95% CI) | P |
|---|---|---|
| R0 resection | | |
| No | 5.31 (2.44–11.52) | **<0.001** |
| Yes | ref | |

HBsAg, hepatitis B surface antigen; HCV, hepatitis C virus; INR, International Normalized Ratio; AFP, Alpha fetoprotein; BCLC, Barcelona Clinic Liver Cancer; BMI, body mass index

the median (IQR) tumor size was 66 (55–90) mm, 66.3% of the patients had microvascular invasion, and 10.9% of the patients had microsatellite nodules. However, only 2.2% of the patients had poor tumor differentiation and only 4.3% of the patients did not undergo R0 resection. All these variables were associated with OS [13].

**Table 3. Risk factors for mortality (multivariate analysis).**

| Variables | HR (95% CI) | P |
|---|---|---|
| Age, years [median (IQR)] | | |
| <= 65 | ref | |
| >65 | 2.14 (1.37–3.33) | **0.001** |
| Bilirubin | | |
| <= 12 | ref | |
| >12 | 1.39 (0.84–2.31) | 0.195 |
| AFP (ng/dL) | | |
| <= 40 | ref | |
| >40 | 1.29 (0.77–2.14) | 0.331 |
| BCLC staging classification | | |
| 0 | 0.78 (0.34–1.78) | 0.558 |
| A | ref | |
| A1 | 0.97 (0.53–1.79) | 0.930 |
| B | 2.10 (1.16–3.78) | **0.014** |
| Type of resection | | |
| Minor | ref | |
| Major | 1.36 (0.84–2.19) | 0.211 |
| Tumor Differentiation | | |
| Well or moderate | ref | |
| Poor | 2.60 (1.26–5.35) | **0.010** |
| Microvascular invasion | | |
| No | ref | |
| Yes | 2.00 (1.21–3.29) | **0.007** |
| Microsatellites nodules | | |
| No | ref | |
| Yes | 3.73 (2.09–6.64) | **<0.001** |
| R0 resection | | |
| No | 1.77 (0.77–4.09) | 0.180 |
| Yes | ref | |

AFP, Alpha fetoprotein; BCLC, Barcelona Clinic Liver Cancer;

Tsilimigras et al. enrolled 157 BCLC-B patients [7]; in contrast, there were only 73 BCLC-B patients enrolled in our study. Furthermore, the BCLC stage B is heterogeneous. In 2012, a panel of experts [14] proposed a classification of BCLC B HCC incorporating the concept of the tumor burden according to the "beyond Milan" and the "within up-to-7" criteria. Other classifications of BCLC stage B were proposed by Kim et al. [15] and the Japanese Society of Transcatheter Hepatic Arterial Embolization [16].

In general, given the debate regarding the prognostic stratification on single large HCC, the discrepancy of data between the present study and the one reported by Tsilimigras et al. [7] could be due to different aetiology or patient characteristics between western and eastern cohorts.

The leading etiology of chronic liver disease in HCCs among Asians (e.g. in Taiwan) is HBV. The leading etiologies of chronic liver disease in HCCs in western countries are HCV, alcohol use disorder, and non-alcoholic fatty liver disease (NAFLD) [17]. Significant proportions of patients with non-cirrhotic liver were noted in the study by Tsilimigras et al. (62.4%) [7] and in our study (58.2%). However, the leading etiology of non-cirrhotic liver in our study was HBV infection. In contrast, the leading etiology of non-cirrhotic liver in the Tsilimigras et al. study may have been NAFLD [7].

Tsilimigras et al. enrolled 814 patients in their study. Among those 814 patients, 253 (31.4%) had HCV infections, 134 (16.5%) had HBV infections, and 508 (62.4%) were non-cirrhotic [7]. Tsilimigras et al. did not mention how many patients in their cohort had NAFLD [7]. However, a recent study from the United States reported that NAFLD was the most common liver disease in patients with non-cirrhotic HCC [17]. As has been mentioned in the EASL guidelines, HCC may occur in non-cirrhotic livers in patients with NAFLD [18–20]. Surgical resections in patients with NAFLD are associated with a significant rate of severe complications, although post-operative mortality has remained low. Obesity related co-morbidities such as lung dysfunction, cardiovascular disease, type 2 diabetes mellitus, hypertension, dyslipidemia, and metabolic syndrome are commonly observed in NAFLD patients and are associated with worse prognosis [6]. Tsilimigras et al. enrolled 448 (55%) patients with a Charlson comorbidity index score >3. There were no significant differences in 3- and 5-year survival between the patients with a Charlson comorbidity index score >3 and those with a score ≤3 (p = 0.102). Furthermore, they enrolled 457 (57.7%) patients with an American Society of Anesthesiologists performance score >2. There were no significant differences in 3- and 5-year survival between the patients with an American Society of Anesthesiologists performance score >2 and those with a score ≤2 (p = 0.430). Tsilimigras et al. did not mention how many of the patients in their study had post-surgical mortality or severe complications [7].

316 (58.2%) of the patients in our cohort were non-cirrhotic. Among those 316 patients, 171 (54.1%) were HBsAg positive, 88 (27.8%) were anti-HCV positive, and 65 (20.6%) were both HBsAg and anti-HCV negative. Among the 65 patients who were both HBsAg and anti-HCV negative, only 4 had alcohol use disorder. In our study, data on post-surgical complications is not included because such data was not available in the HCC registry dataset we used. The etiologies of chronic liver disease listed in our HCC registry data included being HBsAg positive, being anti-HCV positive, and alcohol use disorder, or none of the aforementioned etiologies were present. We defined a habitual drinker as someone with alcohol use disorder in our study. Otherwise, we did not have data indicating how many of the patients had NAFLD.

The limited number of patients with alcohol use disorder in our cohort suggests that the reported daily alcohol consumption of the patients may not have been reliable and could have been underestimated. We did not use screening tools [e.g. quantity frequency questionnaires and diaries or the AUDIT (Alcohol Use Disorders Inventory Test [21])] to identify alcohol use disorders among patients with HCC in our daily practice.

The strength of the present study is the short study period (2011–2017), in which there were not significant changes in surgical techniques, perioperative care, or anesthetic management, all of which influence the patient survival. Second, in the present single center study, the surgeries were all performed by the same highly experienced surgeons, including Yong CC, Wang CC, Chen CL, Lin CC and Liu YW. This should have limited the effects of the operator factor on surgical mortality. There were only 3 (0.6%) perioperative deaths within 90 days of LR in the present cohort. Third, we checked the vital status of these patients by using Cancers Screening and Tracing Information Integrated System, Taiwan (https://hosplab.hpa.gov.tw/CSTIIS/index.aspx). We could make sure the vital status of every single patient enrolled in this study.

There were also some limitations in the present study. First, this is a retrospective study. Second, there were limited numbers of single large HCC and BCLC stage B cases. Third, there was a lack of variables that may be associated with OS in the HCC patients who underwent LR, such as anatomic resection [22] and severe comorbidities. In our hospital, the general principle for LR of HCC followed the recommendations of the EASL guidelines [6], which are based on multi-parametric composite assessment of liver function, the extent of hepatectomy, the future liver remnant, the performance status, and the patient's comorbidities. Therefore, the number of patients with severe comorbidities who underwent LR should have been limited in our cohort. A systematic review and meta-analysis reported that anatomic resection was associated with a decreased risk of death at 5 years (HR: 0.88, 95% CI: 0.79–0.97, p = 0.01) [22]. However, retrospective studies linking anatomic resections and better outcomes should be interpreted with caution. There is a propensity to perform wider LR in patients with well-preserved liver function [6]. Further, some case–control studies using propensity score matching have failed to show this benefit [23–25]. Fourthly, the HCC registry data used in the present study only recorded the first-line therapy.

In conclusion, the results of this study indicated that the prognosis following LR among patients with single large HCC was similar to that among patients with BCLC stage A. The prognosis for single large HCC should thus be considered comparable to that for BCLC stage A.

## Supporting information

**S1 Raw data. Raw data of this cohort.**
(XLSX)

## Acknowledgments

The authors thank Cancer Center, Kaohsiung Chang Gung Memorial Hospital for the provision of HCC registry data and MS Chiang Yi Chen for her administrative assistance with our manuscript. The authors thank Chih-Yun Lin and Nien-Tzu Hsu and the Biostatistics Center, Kaohsiung Chang Gung Memorial Hospital for statistics work.

## Author Contributions

**Conceptualization:** Yi-Hao Yen.

**Data curation:** Yi-Hao Yen.

**Formal analysis:** Yi-Hao Yen.

**Funding acquisition:** Yi-Hao Yen.

**Investigation:** Yi-Hao Yen.

**Methodology:** Yi-Hao Yen.

**Project administration:** Yi-Hao Yen.

**Resources:** Yi-Hao Yen.

**Supervision:** Chih-Che Lin, Chee-Chien Yong, Chih-Chi Wang, Chao-Long Chen, Jing-Houng Wang.

**Writing – original draft:** Yueh-Wei Liu, Yi-Hao Yen.

**Writing – review & editing:** Yi-Hao Yen.

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
