## [Decision Letter · Decision Letter 0]

17 Feb 2020

PONE-D-20-02819

Prognosis after resection of single large hepatocellular carcinoma: results from an Asian high-volume liver surgery center

PLOS ONE

Dear dr Yen,

Thank you for submitting your manuscript to PLOS ONE. After careful consideration, we feel that it has merit but does not fully meet PLOS ONE’s publication criteria as it currently stands. Therefore, we invite you to submit a revised version of the manuscript that addresses the points raised during the review process.

We would appreciate receiving your revised manuscript by Apr 02 2020 11:59PM. To enhance the reproducibility of your results, we recommend that if applicable you deposit your laboratory protocols in protocols.io, where a protocol can be assigned its own identifier (DOI) such that it can be cited independently in the future. For instructions see: http://journals.plos.org/plosone/s/submission-guidelines#loc-laboratory-protocols

We look forward to receiving your revised manuscript.

Kind regards,

Gianfranco D. Alpini

Academic Editor

PLOS ONE

Journal Requirements:

Reviewers' comments:

Reviewer's Responses to Questions

**Comments to the Author**

1. Is the manuscript technically sound, and do the data support the conclusions?

Reviewer #1: Yes

Reviewer #2: Yes

2. Has the statistical analysis been performed appropriately and rigorously? 

Reviewer #1: Yes

Reviewer #2: Yes

3. Have the authors made all data underlying the findings in their manuscript fully available?

Reviewer #1: Yes

Reviewer #2: Yes

4. Is the manuscript presented in an intelligible fashion and written in standard English?

Reviewer #1: Yes

Reviewer #2: Yes

5. Review Comments to the Author

Reviewer #1: This retrospective clinical study focuses on a controversial issue: should SLHCC (>5 cm) be staged as A or B in the BCLC system? It’s very important because different stages indicate different prognosis and furtherly, lead to different treatment plans for HCC patients. The manuscript is well written with scientific methods. The sample size is large and the data from a large liver surgery center is valid and convincing. The conclusion of this research: the prognosis for SLHCC should be considered comparable to that for BCLC stage A, supports the staging method of BCLC system. In the discussion section, the differences among this study and some other previous studies, such as the one published on Annals of Surgical Oncology (Tsilimigras, D.I., Bagante, F., Sahara, K. et al. Ann Surg Oncol (2019) 26: 3693. ) are well illustrated. Therefore, its clinical significance is obvious, especially for Asian patients with HCC.

There are only a few issues that I may need to confirm:

1. What is the exact year when SLHCC was designated as stage A? At the beginning of the introduction, it writes in 2011. But the reference #4 indicates the guideline was published in 2012. It seems conflicted.

2. All the patients enrolled in this research underwent liver resection. Are all the surgeries curable or curative-intent? Are there any palliative operations included?

3. In the discussion section, it writes that the surgeries were all performed by the same highly experienced surgeons, including Yong CC, Wang CC, Chen CL, Lin CC and Liu YW. Do you mean the surgery skill or proficiency is an infecting factor for the different prognosis?

4. In this study, all the data is collected from Asian patients. Do the heterogeneities among Asian population and other ethics make a difference? It’s better to discuss more about it when compare this study with the previous one published on Annals of Surgical Oncology. Although this article is similar to that one in structure, it would make this one more distinctive.

5. There are some details about the statistical methods. For example, when the multivariate analysis for mortality is performed, the results of univariate analysis should be considered. But in this article, all the items included in the univariate analysis seems not derived from multivariate analysis.

6. Since all the statistic methods has been described in the section Statistical Analysis, there is no need to repeat after that. If not, describe it in one section. Please pay attention to the Table 1 on Page 14 and the log rank test on Page 16.

7. The numbers below figure 2 are confusing. The survival curve is clear enough to show the significant differences. What does the number at risk mean? The numbers seem not to match the curve above.

Reviewer #2: In the present manuscript by Liu Y-W and co-workers, authors aimed to evaluate the prognosis of patients with a single large HCC undergoing liver resection. The BCLC classification scheme assigns these patients in stage A; however, a recent study from western countries indicated that prognosis following LR among patients with single large HCC was similar to that among patients presenting with BCLC-B HCCs.

In the present manuscript, authors analysed retrospectively a cohort of 543 patients and their analysis indicated that that the prognosis following LR among patients with single large HCC was similar to that among patients with BCLC stage A.

The study is interesting, and limitations are well acknowledged in the discussion. I have few minor comments.

- I noted that only 21.7% of patients in authors’ cohort were cirrhotic. This is different in comparison with western countries where a higher number of HCC arise on a cirrhosis background. A comparison of this aspect could be included in the discussion. Moreover, authors should provide information on aetiology of underlying liver disease, particularly the presence of alcohol consumption or NAFLD.

- In general, given the debate regarding the prognostic stratification on single large HCC, the discrepancy of data between the present study and the one reported in ref# 7 could be due to different aetiology or patient characteristics between western and eastern cohorts.; this should be acknowledged.

6. PLOS authors have the option to publish the peer review history of their article (what does this mean?). If published, this will include your full peer review and any attached files.

Reviewer #1: No

Reviewer #2: No

---

## [Author Response · Author response to Decision Letter 0]

6 Mar 2020

Reviewer #1: This retrospective clinical study focuses on a controversial issue: should SLHCC (>5 cm) be staged as A or B in the BCLC system? It’s very important because different stages indicate different prognosis and furtherly, lead to different treatment plans for HCC patients. The manuscript is well written with scientific methods. The sample size is large and the data from a large liver surgery center is valid and convincing. The conclusion of this research: the prognosis for SLHCC should be considered comparable to that for BCLC stage A, supports the staging method of BCLC system. In the discussion section, the differences among this study and some other previous studies, such as the one published on Annals of Surgical Oncology (Tsilimigras, D.I., Bagante, F., Sahara, K. et al. Ann Surg Oncol (2019) 26: 3693. ) are well illustrated. Therefore, its clinical significance is obvious, especially for Asian patients with HCC.

There are only a few issues that I may need to confirm:

1. What is the exact year when SLHCC was designated as stage A? At the beginning of the introduction, it writes in 2011. But the reference #4 indicates the guideline was published in 2012. It seems conflicted.

Response: Thank you so much for your comments. I have corrected the error in the introduction; it was 2012. Please page 3, 1st paragraph, line 1 and page 5, 2nd paragraph, line 3.

2. All the patients enrolled in this research underwent liver resection. Are all the surgeries curable or curative-intent? Are there any palliative operations included?

Response: Thank you again for your comments. All the surgeries were curable or curative-intent. There were no palliative operations included in this study. please see page 3, methods and page 6, 3rd paragraph, line 3. 

3. In the discussion section, it writes that the surgeries were all performed by the same highly experienced surgeons, including Yong CC, Wang CC, Chen CL, Lin CC and Liu YW. Do you mean the surgery skill or proficiency is an infecting factor for the different prognosis?

Response: Thank you again for your comments. There were only 3 (0.6%) perioperative deaths within 90 days of surgery in the present cohort. Theoretically, the surgery skill or proficiency could be an infecting factor for perioperative mortality. However, we did not find any references regarding this issue. 

4. In this study, all the data is collected from Asian patients. Do the heterogeneities among Asian population and other ethics make a difference? It’s better to discuss more about it when compare this study with the previous one published on Annals of Surgical Oncology. Although this article is similar to that one in structure, it would make this one more distinctive.

Response: Thank you again for your comments. 

The leading etiology of chronic liver disease in HCCs among Asians (e.g. in Taiwan) is HBV. The leading etiologies of chronic liver disease in HCCs in western countries are HCV, alcohol use disorder, and non-alcoholic fatty liver disease (NAFLD) [17]. Significant proportions of patients with non-cirrhotic liver were noted in the study by Tsilimigras et al. (62.4%) [7] and in our study (58.2%). However, the leading etiology of non-cirrhotic liver in our study was HBV infection. In contrast, the leading etiology of non-cirrhotic liver in the Tsilimigras et al. study may have been NAFLD [7].

Tsilimigras et al. enrolled 814 patients in their study. Among those 814 patients, 253 (31.4%) had HCV infections, 134 (16.5%) had HBV infections, and 508 (62.4%) were non-cirrhotic [7]. Tsilimigras et al. did not mention how many patients in their cohort had NAFLD [7]. However, a recent study from the United States reported that NAFLD was the most common liver disease in patients with non-cirrhotic HCC [17]. As has been mentioned in the EASL guidelines, HCC may occur in non-cirrhotic livers in patients with NAFLD [18-20]. Surgical resections in patients with NAFLD are associated with a significant rate of severe complications, although post-operative mortality has remained low. Obesity related co-morbidities such as lung dysfunction, cardiovascular disease, type 2 diabetes mellitus, hypertension, dyslipidemia, and metabolic syndrome are commonly observed in NAFLD patients and are associated with worse prognosis [6]. Tsilimigras et al. enrolled 448 (55%) patients with a Charlson comorbidity index score >3. There were no significant differences in 3- and 5-year survival between the patients with a Charlson comorbidity index score >3 and those with a score ≤3 (p= 0.102). Furthermore, they enrolled 457 (57.7%) patients with an American Society of Anesthesiologists performance score >2. There were no significant differences in 3- and 5-year survival between the patients with an American Society of Anesthesiologists performance score >2 and those with a score ≤2 (p=0.430). Tsilimigras et al. did not mention how many of the patients in their study had post-surgical mortality or severe complications [7].

316 (58.2 %) of the patients in our cohort were non-cirrhotic. Among those 316 patients, 171 (54.1%) were HBsAg positive, 88 (27.8%) were anti-HCV positive, and 65 (20.6%) were both HBsAg and anti-HCV negative. Among the 65 patients who were both HBsAg and anti-HCV negative, only 4 had alcohol use disorder. In our study, data on post-surgical complications is not included because such data was not available in the HCC registry dataset we used. The etiologies of chronic liver disease listed in our HCC registry data included being HBsAg positive, being anti-HCV positive, and alcohol use disorder, or none of the aforementioned etiologies were present. We defined a habitual drinker as someone with alcohol use disorder in our study. Otherwise, we did not have data indicating how many of the patients had NAFLD. Please see page 25-28. 

5. There are some details about the statistical methods. For example, when the multivariate analysis for mortality is performed, the results of univariate analysis should be considered. But in this article, all the items included in the univariate analysis seems not derived from multivariate analysis.

Response: Thank you again for your comments. We have changed the multivariate method used. The variables with P values less than 0.05 in the univariate analysis were included in the multivariate model, and the result showed that no difference in overall survival was noted for the patients with BCLC stage A versus A1 after adjusting for competing factors (hazard ratio= 0.97, 95% confidence interval= 0.53-1.79; p = 0.93) (Table 3). Please see page 3, last 2 lines, page 8, 1st paragraph, line 6,7 and page 20. 

6. Since all the statistic methods has been described in the section Statistical Analysis, there is no need to repeat after that. If not, describe it in one section. Please pay attention to the Table 1 on Page 14 and the log rank test on Page 16.

Response: Thank you again for your comments. I have corrected the relevant text accordingly. 

7. The numbers below figure 2 are confusing. The survival curve is clear enough to show the significant differences. What does the number at risk mean? The numbers seem not to match the curve above.

Response: Thank you again for your comments. I have deleted the number at risk to avoid causing confusion. The number at risk was meant to indicate the number of patients who were still being followed up in our hospital. For example, the number at risk with BCLC stage 0 was 29 at 60 months and 19 at 72 months. However, the overall survival probability curve of BCLC stage 0 was the same during this period. Therefore, the decrease in the number at risk indicated that 10 of the 29 patients were still alive but were lost to follow-up after 60 months (right-censored). That could explain why the number at risk was progressively decreasing even as the overall survival probability was relatively high – that is, most of the patients who were not followed up in our hospital were still alive

Reviewer #2: In the present manuscript by Liu Y-W and co-workers, authors aimed to evaluate the prognosis of patients with a single large HCC undergoing liver resection. The BCLC classification scheme assigns these patients in stage A; however, a recent study from western countries indicated that prognosis following LR among patients with single large HCC was similar to that among patients presenting with BCLC-B HCCs. In the present manuscript, authors analysed retrospectively a cohort of 543 patients and their analysis indicated that that the prognosis following LR among patients with single large HCC was similar to that among patients with BCLC stage A. The study is interesting, and limitations are well acknowledged in the discussion. I have few minor comments.

- I noted that only 21.7% of patients in authors’ cohort were cirrhotic. This is different in comparison with western countries where a higher number of HCC arise on a cirrhosis background. A comparison of this aspect could be included in the discussion. Moreover, authors should provide information on aetiology of underlying liver disease, particularly the presence of alcohol consumption or NAFLD.

Response: Thank you so much for your comments. In our study, 20 (21.7%) of the patients with a single large HCC were cirrhotic. Among the entire cohort, 227 (41.8 %) patients were cirrhotic, and 316 (58.2 %) were non-cirrhotic. 

The leading etiology of chronic liver disease in HCCs among Asians (e.g. in Taiwan) is HBV. The leading etiologies of chronic liver disease in HCCs in western countries are HCV, alcohol use disorder, and non-alcoholic fatty liver disease (NAFLD) [17]. Significant proportions of patients with non-cirrhotic liver were noted in the study by Tsilimigras et al. (62.4%) [7] and in our study (58.2%). However, the leading etiology of non-cirrhotic liver in our study was HBV infection. In contrast, the leading etiology of non-cirrhotic liver in the Tsilimigras et al. study may have been NAFLD [7].

Tsilimigras et al. enrolled 814 patients in their study. Among those 814 patients, 253 (31.4%) had HCV infections, 134 (16.5%) had HBV infections, and 508 (62.4%) were non-cirrhotic [7]. Tsilimigras et al. did not mention how many patients in their cohort had NAFLD [7]. However, a recent study from the United States reported that NAFLD was the most common liver disease in patients with non-cirrhotic HCC [17]. As has been mentioned in the EASL guidelines, HCC may occur in non-cirrhotic livers in patients with NAFLD [18-20]. Surgical resections in patients with NAFLD are associated with a significant rate of severe complications, although post-operative mortality has remained low. Obesity related co-morbidities such as lung dysfunction, cardiovascular disease, type 2 diabetes mellitus, hypertension, dyslipidemia, and metabolic syndrome are commonly observed in NAFLD patients and are associated with worse prognosis [6]. Tsilimigras et al. enrolled 448 (55%) patients with a Charlson comorbidity index score >3. There were no significant differences in 3- and 5-year survival between the patients with a Charlson comorbidity index score >3 and those with a score ≤3 (p= 0.102). Furthermore, they enrolled 457 (57.7%) patients with an American Society of Anesthesiologists performance score >2. There were no significant differences in 3- and 5-year survival between the patients with an American Society of Anesthesiologists performance score >2 and those with a score ≤2 (p=0.430). Tsilimigras et al. did not mention how many of the patients in their study had post-surgical mortality or severe complications [7].

316 (58.2 %) of the patients in our cohort were non-cirrhotic. Among those 316 patients, 171 (54.1%) were HBsAg positive, 88 (27.8%) were anti-HCV positive, and 65 (20.6%) were both HBsAg and anti-HCV negative. Among the 65 patients who were both HBsAg and anti-HCV negative, only 4 had alcohol use disorder. In our study, data on post-surgical complications is not included because such data was not available in the HCC registry dataset we used. The etiologies of chronic liver disease listed in our HCC registry data included being HBsAg positive, being anti-HCV positive, and alcohol use disorder, or none of the aforementioned etiologies were present. We defined a habitual drinker as someone with alcohol use disorder in our study. Otherwise, we did not have data indicating how many of the patients had NAFLD. 

The limited number of patients with alcohol use disorder in our cohort suggests that the reported daily alcohol consumption of the patients may not have been reliable and could have been underestimated. We did not use screening tools [e.g. quantity frequency questionnaires and diaries or the AUDIT (Alcohol Use Disorders Inventory Test [21])] to identify alcohol use disorders among patients with HCC in our daily practice. Please page 25-28 and Table 1. 

- In general, given the debate regarding the prognostic stratification on single large HCC, the discrepancy of data between the present study and the one reported in ref# 7 could be due to different aetiology or patient characteristics between western and eastern cohorts.; this should be acknowledged.

 Response: Thank you again for your comments. I have acknowledged this important point of view in the revised text. Please see page 25, 2nd paragraph.

---

## [Decision Letter · Decision Letter 1]

12 Mar 2020

Prognosis after resection of single large hepatocellular carcinoma:results from an Asian high-volume liver surgery center

PONE-D-20-02819R1

Dear Dr. Yi-Hao Yen,

We are pleased to inform you that your manuscript has been judged scientifically suitable for publication and will be formally accepted for publication once it complies with all outstanding technical requirements.

With kind regards,

Gianfranco D. Alpini

Academic Editor

PLOS ONE

Additional Editor Comments (optional):

Reviewers' comments:

Reviewer's Responses to Questions

**Comments to the Author**

1. If the authors have adequately addressed your comments raised in a previous round of review and you feel that this manuscript is now acceptable for publication, you may indicate that here to bypass the “Comments to the Author” section, enter your conflict of interest statement in the “Confidential to Editor” section, and submit your "Accept" recommendation.

Reviewer #1: All comments have been addressed

Reviewer #2: All comments have been addressed

2. Is the manuscript technically sound, and do the data support the conclusions?

Reviewer #1: Yes

Reviewer #2: Yes

3. Has the statistical analysis been performed appropriately and rigorously? 

Reviewer #1: Yes

Reviewer #2: Yes

4. Have the authors made all data underlying the findings in their manuscript fully available?

Reviewer #1: Yes

Reviewer #2: Yes

5. Is the manuscript presented in an intelligible fashion and written in standard English?

Reviewer #1: Yes

Reviewer #2: Yes

6. Review Comments to the Author

Reviewer #1: (No Response)

Reviewer #2: (No Response)

7. PLOS authors have the option to publish the peer review history of their article (what does this mean?). If published, this will include your full peer review and any attached files.

Reviewer #1: No

Reviewer #2: No

---

## [Editor Report · Acceptance letter]

16 Mar 2020

PONE-D-20-02819R1 

Prognosis after resection of single large hepatocellular carcinoma: results from an Asian high-volume liver surgery center 

Dear Dr. Yen:

I am pleased to inform you that your manuscript has been deemed suitable for publication in PLOS ONE. Congratulations! Your manuscript is now with our production department. 

With kind regards,

on behalf of

Dr. Gianfranco D. Alpini 

Academic Editor

PLOS ONE